# Compromised Biomechanical Properties, Cell–Cell Adhesion and Nanotubes Communication in Cardiac Fibroblasts Carrying the Lamin A/C D192G Mutation

**DOI:** 10.3390/ijms22179193

**Published:** 2021-08-25

**Authors:** Veronique Lachaize, Brisa Peña, Catalin Ciubotaru, Dan Cojoc, Suet Nee Chen, Matthew R. G. Taylor, Luisa Mestroni, Orfeo Sbaizero

**Affiliations:** 1Department of Engineering and Architecture, University of Trieste, Via Valerio 10, 34127 Trieste, Italy; lachaize.v@gmail.com; 2CU-Cardiovascular Institute, University of Colorado Anschutz Medical Campus, 12700 E. 19th Ave., Aurora, CO 80045, USA; brisa.penacastellanos@cuanschutz.edu (B.P.); suet.chen@cuanschutz.edu (S.N.C.); matthew.taylor@cuanschutz.edu (M.R.G.T.); luisa.mestroni@cuanschutz.edu (L.M.); 3Consortium for Fibrosis Research & Translation, Anschutz Medical Campus, University of Colorado, 12700 E. 19th Ave., Aurora, CO 80045, USA; 4Bioengineering Department, University of Colorado Denver Anschutz Medical Campus, Bioscience 2 1270 E. Montview Ave., Suite 100, Aurora, CO 80045, USA; 5Institute of Materials, National Research Council of Italy (CNR_IOM), Area Science Park Basovizza, 34149 Trieste, Italy; ciubotaru@iom.cnr.it (C.C.); cojoc@iom.cnr.it (D.C.)

**Keywords:** arrhythmogenic cardiomyopathy (ACM), lamin A/C, atomic force microscopy (AFM), cell–cell adhesion, neonatal rat ventricular fibroblasts (NRVF), tunneling nanotubes (TNT)

## Abstract

Clinical effects induced by arrhythmogenic cardiomyopathy (ACM) originate from a large spectrum of genetic variations, including the missense mutation of the lamin A/C gene (*LMNA*), *LMNA* D192G. The aim of our study was to investigate the biophysical and biomechanical impact of the *LMNA* D192G mutation on neonatal rat ventricular fibroblasts (NRVF). The main findings in mutated NRVFs were: (i) cytoskeleton disorganization (actin and intermediate filaments); (ii) decreased elasticity of NRVFs; (iii) altered cell–cell adhesion properties, that highlighted a strong effect on cellular communication, in particular on tunneling nanotubes (TNTs). In mutant-expressing fibroblasts, these nanotubes were weakened with altered mechanical properties as shown by atomic force microscopy (AFM) and optical tweezers. These outcomes complement prior investigations on *LMNA* mutant cardiomyocytes and suggest that the *LMNA* D192G mutation impacts the biomechanical properties of both cardiomyocytes and cardiac fibroblasts. These observations could explain how this mutation influences cardiac biomechanical pathology and the severity of ACM in *LMNA*-cardiomyopathy.

## 1. Introduction

Arrhythmogenic cardiomyopathy (ACM) is a myocardial disease characterized by a high risk of life-threatening ventricular arrhythmias, sudden cardiac death, and progression towards heart failure [1,2]. This disease, which can predominantly affect the right ventricle (arrhythmogenic right ventricular cardiomyopathy or ARVC), the left ventricle (arrhythmogenic left ventricular cardiomyopathy (ALVC), or arrhythmogenic dilated cardiomyopathy (DCM)), or both (biventricular ACM) is generated by a large spectrum of genes including the lamin A/C gene (*LMNA*) [1]. Many ACM genes, such as *LMNA* [3], induce important alterations of the signal transduction and the intercellular communication mechanism in cardiac cells [4,5,6]. In 2015, Lanzicher et al. investigated the biomechanical impact of *LMNA* missense mutations in neonatal rat ventricular myocytes (NRVM), and found damages on the actin cytoskeleton, suggesting loss of actin filament anchorage by dysfunction of the laminar proteins at the nuclear membrane as the initiating alteration [7,8]. Their results were further confirmed by different studies, in which it was demonstrated that lamina network mutations induce an alteration of actin filaments, microtubule, and intermediate filament anchorage in the nuclear membrane, a region critical for cytoskeleton organization [9,10]. Moreover, the link between the lamina network and the cytoskeleton filaments is ensured by a protein complex: LInker of Nucleoskeleton and Cytoskeleton (LINC). A functional nuclear lamin cortex is therefore necessary for the cytoskeleton anchorage, at the nuclear membrane LINC complex, and for maintaining the cytoskeleton integrity [11,12]. As previously reported, atomic force microscopy (AFM) studies highlighted that cardiomyocytes expressing a LMNA D192G mutation have alterations of membrane biomechanical properties (stiffness and fragility), due to cytoskeletal structural modifications, and decreased adhesion between the cell membrane and the AFM probe [7,13]. Furthermore, LMNA mutations appear to disrupt cell adhesion, potentially hampering the communication between cardiomyocytes [7]. However, the link between the defective lamina network, defective cell adhesion, and the ACM phenotype has not been demonstrated. 

In 2004, A. Rustom et al. reported a cell–cell communication system which offers a new perspective: cellular communication by tunneling nanotubes (TNT). In their work, this communication works as a “highway” connecting two cells and it is composed by nanoscale membrane tubes with diameters between 50 and 200 nm, containing actin filaments and allowing the exchange of biological material (such as organelles, plasma membrane components, and cytoplasmic small particles [14] like mitochondria [15], HLA proteins [16], viruses [17], lysosomes [18], and calcium [16]) (Figure 1A).

In particular conditions (such as cell rescue phenomena), microtubules can be found in TNTs, but in typical conditions, TNTs are composed mainly of actin filaments [19,20,21]. In the literature, different studies assessed the impact of this communication mechanism in physio/pathology condition, like the infection of HIV [17] and cancer cell proliferation [18,22,23]. In in vivo conditions, TNT formation uses two different mechanisms: the filopodia production, followed with lamellipodia and, finally, the TNT between two cells (Figure 1B), or by direct contact between two cells (Figure 1C). Some studies suggest that the interaction of two connexin 43 (Cx43) proteins can initiate TNT formation [14,24,25]. Cx43 interactions stabilize the connection between two cells, allowing the cytoskeleton filament of each cell to organize and form TNTs. Recently, K. He et al. not only demonstrated the presence of TNT in the cardiac tissue between two cardiomyocytes, but also between cardiac fibroblasts and cardiomyocytes [26]. These data suggest that the presence of TNT between fibroblasts and cardiomyocytes may be important in cellular crosstalk. Indeed, cardiac fibroblasts have an essential role: they are the largest cell populations in the heart, followed by cardiomyocytes, and cardiac fibroblasts contribute to the structural, biochemical, mechanical, and electrical properties of the myocardium tissue [27,28], modulation of the myocardial response [29], and production of the extra cellular matrix (ECM) [30]. If chronically stimulated they may induce fibrosis [31]. Fibroblasts could therefore have an important role in ACM. Indeed, the multiple functions of fibroblasts could be altered by TNT damages. Decrease of communication by TNT dysfunction could impact the modulation of the myocardial response to stress and trigger the production of the extracellular matrix. Therefore, the aim of our study was to investigate the impact of an *LMNA* mutation on cardiac fibroblasts. To this purpose, biophysical and biomechanical measurements were carried out on living neonatal rat ventricular fibroblasts (NRVF) expressing the LMNA D192G mutation. This mutation was selected because it represents a mutation associated with disruption of nuclear envelope morphology [32]. It has been described in one family with a severe phenotype where symptom onset and death/transplantation happened at 30.5 (±6.4) and 32.0 (±7.1) years, respectively: both patients died before the age of 40 years [33]. Although the small number of observations of this rare mutation must be taken into consideration, clinical data suggest that LMNA D192G have a very severe outcome.

We used a multidisciplinary approach including cell–cell adhesion measurements by cell spectroscopy using atomic force microscopy [34]. The cell–cell adhesion tests identified differences in adhesion patterns such as energy, maximum adhesion, rupture, and nanotube characteristics between two living NRVFs [35,36]. Moreover, we analyzed adhesion parameters in wild type (WT) NRVFs after blocking Cx43 to understand the role of this protein in the TNT. Furthermore, we studied the nanotubes’ mechanical properties by optical tweezers (OT) and their structure by fluorescence staining experiments. OT measurements allowed us to quantify forces involved in the extrusion of only one nanotube under different conditions (mutated or WT). Here, we report how complementary and multidisciplinary biophysical and biological tools generated a new mechanistic insight into the impact of the LMNA D192G mutation on the fibroblast structure and function.

## 2. Results

Since adenovirus causes a transient transfection, to control if our cells were still expressing the episome, we checked them, during and after the experiment, using GFP fluorescence.

### 2.1. Immunofluorescence Showed Actin Disorganization and Brittle TNTs in Fibro-MT

Neonatal rat ventricular fibroblasts (NRVFs) were prepared as (i) non-treated, transfection control NRVFs (Fibro-CT), as (ii) wild type NRVFs (Fibro-WT), and as (iii) NRVFs with LMNA D192G mutation (Fibro-MT). For each condition, actin filaments and intermediate filaments were labeled by immunofluorescence, in order to observe the effect of LMNA D192G mutation on their cytoskeleton. Fibro-MT showed actin network damage which stands out as disorganization of the cytoskeleton when compared with both Fibro-WT and Fibro-CT (Figure 2A).

As far as the TNTs are concerned, when they were produced by Fibro-MT they displayed a “brittle” morphology with a thickness of 0.66 ± 0.36 µm (Figure 2B), while TNTs produced by Fibro-WT and Fibro-CT were thicker: 2.55 ± 1.29 µm and 2.63 ± 1.48 µm respectively.

In all three conditions actin and intermediate filaments were present inside the TNTs (Figure 2). When TNTs linked two cells, they could be as long as 278 ± 23 nm, but this length was greatly reduced in Fibro-MT (70 ± 11 nm). Our TNT observations showed therefore that the morphological damages in both the actin network and the intermediate filament in the LMNA D192G mutant fibroblasts also influence the TNTs structure making them thinner, shorter, and more brittle.

The cylindrical shape of membrane nanotubes results from a compromise between surface tension and bending rigidity (1). Taking the analogy between tethers and filopodia as a guide, the force that the cell membrane exerts on the cytoskeleton at the ends of the TNT can be calculated if we idealize the TNT as an actin bundle cylinder of radius *r*. In this case, the force *Fmem* exerted on this cylinder by the membrane is equal to the product of the surface tension *N* of the membrane and the circumference of the cylinder:*F*_*mem*_ = 2π*r**N*(1)

This force does not consider membrane bending and/or micro-membrane structures such as invaginations and protrusion deformation, both of which could provide further resistance to TNT extraction, but it is a good approximation that allows us to compare the optical tweezer (OT) results.

As far as the TNT mechanical properties are concerned (2), we can consider it as a beam of length *L* with one fixed end, and with an axial force *F* applied at the free end. The beam will buckle if *F* exceeds the buckling force
*F_buckle_* = π^2^*E I*/4 *L*^2^(2)
where *E* is the Youngs modulus (elasticity) of the beam, and *I* is its moment of inertia.

The maximum TNT length before buckling occurs can be estimated assuming two contrasting hypotheses: (i) actin without cross-linkers inside the TNT (3) and (ii) highly cross-linked actin. In the latter hypothesis, the filaments can be seen as a single unit of effective radius *rbundle* (4).

The first hypothesis, actin without cross-linkers, produces a maximum length before buckling (*L_N-CL_*):(3)$$\;\;\;\;\;\;\;\;L_{N-CL}=\sqrt{\frac{\pi^{2}\;\;\;\frac{\pi \;\;r_{Ac}^{4}}{4}\;\;E_{Ac\;\;}}{4\;\;\frac{F_{mem}}{n}}\;}$$
while highly cross-linked actin will have a maximum length (*L_CL_*):(4)$$L_{CL}=\sqrt{\frac{\pi^{2}\;\;\;\frac{\pi \;\;{\left(\sqrt{\pi \;r_{Ac}}\right)}^{4}}{4}\;\;E_{Ac\;\;}}{4\;F_{mem}}\;}$$
where *E*_*AC*_ is the actin Young’s modulus and *r*_*AC*_ is the actin filament radius, respectively. Therefore, the ratio of (*L_CL_*)/(*L_N-CL_*) is $\sqrt{n}$. This latter result leads to the conclusion that the highest the number of filaments inside each TNT, the more distinct the dissimilarity in the maximum lengths between the cross-linked and uncross-linked TNT. Basically, if the actin inside the TNT is well organized and highly cross-linked, the TNT is more stable mechanically, and can develop much longer. The reduction of the slope of the tether force in Fibro-MT is consistent with two concomitant factors, (i) in order to form tethers, the membrane-cytoskeleton links must be ruptured, (ii) the much smaller contribution from the cytoskeleton remnants left inside the tether itself.

### 2.2. AFM Tests Highlighted Biomechanical Changes in Fibro-MT

In order to support and quantify these initial microscopic observations, measurements of the fibroblasts’ biomechanical properties were also carried out using AFM. Cell elasticity (Young Modulus) was estimated using the Hertz–Sneddon model after collecting force curves obtained by force spectroscopy. Preliminary assessment of these two conditions, morphological and biomechanical comparison between Fibro-CT and Fibro-WT, namely NRVFs infected by the same protocol used for the mutated condition but with an empty plasmid, were also performed to check the adenovirus infection’s impact on NRVFs. Fibro-CT and Fibro-WT shows no significant differences, regarding mechanical properties (using AFM), cell morphology and dimensions (using fluorescence labeling) (Appendix A). Figure 3A shows the Young’s modulus for Fibro-CT, Fibro-WT, and Fibro-MT.

The Young’s modulus values were obtained from the AFM force–deformation data with a cell deformation of 10%, since this range is considered typical of a linear elastic deformation. The elasticity mean value for Fibro-CT is 7.33 ± 0.88 kPa, for Fibro-WT is 7.77 ± 1.21 kPa, while for Fibro-MT the elasticity mean value is reduced to 3.63 ± 0.65 kPa. The elasticity difference between these different conditions proves that the resistance needed to compress the cell with the AFM tip is two times lower for the mutated condition. These differences in elasticity could be explained by the presence of cytoskeleton damages as previously observed in the mutated condition by immunofluorescent labeling (Figure 2). Indeed, Lanzicher et al. already detected this type of damage on the cardiomyocyte actin network expressing LMNA D192G mutation with a lower actin fibers density in the cytoskeleton and the presence of blebs on nuclear membrane [7].

### 2.3. AFM on Cell–Cell Adhesion Confirmed Changes in Fibro-MT

In addition to these results, an innovative approach has been explored: the cell–cell adhesion measurements conducted by AFM. This method allows us to generate nanotubes between two cells and measure the cell walls’ adhesive properties (Figure 4A). As previously mentioned, in the AFM description (Materials and Methods), one cell has been immobilized on a tipless cantilever and put in contact (for 25 s) with a second cell, immobilized in a fibronectin-coated support. During a retraction step, described as the separation between these two cells, one force curve was acquired. By using these retract curves, several adhesion parameters could be quantified, as previously mentioned (Figure 4B): (i) the energy defined as the total energy applied to separate two cells and given by the area under each retract curves; (ii) the maximum force i.e., the maximum value of the force needed to separate the cells; (iii) the rupture pattern which provides information about separation phenomena at short distances; (iv) the nanotube patterns generated during the separation of two cells. In order to analyze the mutation’s overall incidence on adhesive parameters between two NRVFs, energy, and maximum force parameters between Fibro-WT and Fibro-MT were compared. Figure 3B shows boxplots which highlight differences between these cell lines. In the Fibro-CT and Fibro-MT, the separation of the two fibroblasts required less energy compared to Fibro-WT, in particular: for Fibro-CT 32.26 ± 6.38 × 10^−16^ J, for Fibro-WT 34.64 ± 6.47 × 10^−16^ J, and for Fibro-WT 13.76 ± 2.35 × 10^−16^ J, respectively. Furthermore, the resistance to separate the cells in the Fibro-CT and Fibro-WT is stronger: this could be explained if the cell surface generates more links/connections, and it has more affinity with the second cell. The maximum force was indeed 407.97 ± 173.14 pN for Fibro-CT and 431.12 ± 107.1 pN for Fibro-WT, but only 332.54 ± 196.43 pN for Fibro-MT. A force ≈29% lower was therefore enough to separate two mutated NRVFs. In summary, Figure 5 shows the decrease of the compression resistance (as elasticity, Figure 3A) and the decrease of the interaction between cells (by adhesion, Figure 3B,C) for the Fibro-MT.

### 2.4. Force Rupture Distribution and Nanotube Patterns Are Different in Fibro-MT

For a better understanding, rupture and nanotube patterns were analyzed to obtain more information on the phenomena that occur during the two cells’ separation at both short and long distance. Rupture patterns provide information on protein interaction during the first range of separation, defined as withdrawal from 0 to 5 µm. Nanotube patterns characterize the nanotubes formed between two cells and their following break in the second separation range, defined as withdrawal from 5 to 80 µm. The breaking force showed a slight decrease in the mutated condition, but even though there is this trend, this decrease was not statistically significant (Figure 5A). The mean of the rupture pattern for the breaking force is 32.23 ± 19.40 pN for Fibro-CT, 28.98 ± 12.60 pN for Fibro-WT, and 26.7 ± 13.74 pN for Fibro-MT, respectively. Even if the number of rupture pattern analysis is important (cf. Appendix A), the standard deviation is too heterogeneous to highlight a trend.

However, both the energy and maximum force, previously discussed, showed that the adhesion decreases significantly in the mutated conditions and therefore a detailed rupture breaking force distribution was investigated to understand if there were differences in the adhesion proteins involved or in their density when the mutated cells are tested (Figure 3). Therefore, the analysis of breaking force distribution might allow us to get better information on the standard deviation of the mean breaking force. The distribution of the breaking force is shown in (Figure 5B–E) and it displays a multimodal distribution. Peaks are highlighted by fitting with Gaussian function where each peak of the curve represent the interaction break of a specific adhesion protein population [36]. Even though some sub-peaks may be hidden, this assumption allows us to compare the behavior of the different cell lines. Fibro-CT and Fibro-WT show four and five populations of breaking force after Gaussian deconvolutions, respectively. For Fibro-WT the first at 7.8 pN, the second at 17.45, then 36.1 pN, 59.1, and 83.5 pN (Figure 5C). The breaking force distribution for Fibro-MT identified only 3 peaks, the first at 13.1 pN, the second at 28.1 pN, and the last one at 85.2 pN. A fourth peak around 20 pN is present, but the signal is perturbed by the adjacent peaks’ spread (Figure 5D). In both conditions, the same three populations of adhesion protein were identified. The first adhesion protein population presents a breaking force value around 10 pN, the second around 30 pN, and the third around 80 pN. However, both the Fibro-CT and the Fibro-WT present another population around 60 pN, which is not observed in the mutated condition. As previously reported, Cx43 is an adhesion protein that makes the TNTs’ formation possible [14,24,25], we postulate that in Fibro-MT the function of this protein is altered, and could be responsible for the missing population around 60 pN.

### 2.5. Cx43 Is Correlated to the Biomechanical Changes in Fibro-MT

To verify the aforementioned hypothesis, a mimetic protein of Cx43, Gap 27, was used to block the link between two Cx43 and subsequently, the TNTs’ formation. In previous work [10], we examined the levels and localization of Cx43 in cells with LMNA D192G mutation, since Cx43 protein is important for ensuring coordinated contractile action and its amount and localization might be one of the leading mechanisms for arrhythmias. We found that in the presence of this mutation, there was an altered Cx43 localization even though its overall cellular levels were unchanged. Prior to the cell–cell adhesion measurement, Fibro-WT were incubated with Gap27 solution for 1 h. In Figure 5A, the mean of rupture pattern for the breaking force is 7.35 ± 4.68 pN for NRVFs WT incubated with Gap27. Furthermore, Figure 5E shows that the distribution of the breaking forces, and the deconvolution by Gaussian, presents three populations: the first and the highest at 3.4 pN, the second peak at 9.3 pN, and the third at 33.5 pN. Moreover, in all the studied conditions, the adhesion population has a part of breaking forces lower than 10 pN, that could be associated with unspecific interactions [35,37]. All conditions present a peak around 30 pN, which could be one specific adhesion protein population such as integrin. Integrins are cell surface receptors that are instrumental in mediating cell–matrix interactions in all cells, and specifically in fibroblast with its matrix remodeling functions [38]. Few articles present some adhesion measurements of specific protein such as integrin α_5_β_1_ on fibronectin support. In particular, Z. Sun et al. report that the breaking force between integrin α_5_β_1_ and fibronectin is around 40 pN [39]. The value of the breaking force between two integrins in in vivo condition has not been measured yet, but the 30 pN peak might probably be inferred as the integrin population. The M. Horton’s team studied, using the AFM, the breaking force of ligand–receptor interaction [40], and found that the interaction between integrin and RGD-peptide is around 30 to 120 pN. However, they highlighted that the force interaction between integrins is difficult to assess with accuracy. As explained in the article, the affinity of the receptor for a given ligand depends not only on the integrin type but also on conformational changes in the receptor and hence its “activation” status. Moreover, working with living cells leads to a reduced accuracy in measurements due to membrane deformability, ECM, etc.

All conditions present a range of breaking forces around 80 pN, which could be another type of adhesion protein population with a strong interaction force and low frequency on the membrane surface. What is remarkable is that in the Fibro-CT and Fibro-WT conditions, one peak is always present at around 60 pN and this population is lacking after the Gap 27 blocking process. This breaking force likely represents the Cx43 population as previously suggested.

### 2.6. In Fibro-MT, a Decrease of the Highest Value of the TNT’s Breaking Force Was Found

The average of nanotube breaking force presented no significant difference between the CT and WT and the difference between the WT and the MT is small, but in any case, statistically significant: Fibro-CT showed an average force value of 36.71 ± 4.54 pN, Fibro-WT showed a force value of 39.53 ± 4.90 pN, and Fibro-MT a force of 33.61 ± 5.59 pN (Figure 6A).

Additionally, the distribution of nanotube breaking forces was also analyzed. As previously mentioned in the energy and maximum force parameters study, a significant decrease of the cell adhesion in the mutant condition was observed (Figure 3). However, the nanotube breaking-force distribution (Figure 6B–E) presents different multimodal force distribution for the three conditions studied. Each multimodal distribution peak represents a specific nanotube population generated during the cell–cell separation. In the Fibro-WT condition, three populations of nanotubes were recognized: the first population presents a breaking force at 11.1 pN, the second at 31.7, and the third at 58.0 pN. In the Fibro-MT, a similar trend was observed: three populations were underscored, the first at 12.4 pN, the second at 28.2 pN, and the third at 43.3 pN. Even though both conditions present two similar populations of nanotubes, the first is around 10 pN and the second is around 30 pN, in the Fibro-WT, the third population presents a breaking force at 58.0 pN, 34% higher compared to the third population in the Fibro-MT. This decrease of the highest value was also found in the Fibro-WT condition after Gap 27 blocking. Two hypotheses could be suggested to explain this observation: (i) a model where periodicity could suggest that the peak around 10 pN corresponds to the value of the breaking force for a single nanotube, the second peak around 30 pN corresponds to the simultaneous breaking force of two nanotubes, where the third corresponds to three nanotubes, etc.; (ii) a model where each peak corresponds to a specific type of nanotube. The 10 pN peak could be a nanotube composed only by the lipid bilayer of the plasma membrane. The population with a high nanotube breaking force around 60 pN could be a TNT with Cx43 anchor function. The intermediate value could be a hybrid TNT, with the presence of Cx43. In the mutated condition, the lower values of adhesion energy and maximum force due to decrease on Cx43 on the cell membrane might represent their difficulty to initiate the TNT formation.

### 2.7. Optical Tweezers

Optical tweezer (OT) observations were carried out in parallel to the AFM experiments to collect a new range of information about the nanotubes’ pattern (Figure 7A). OT allows us to measure the mechanical properties of a single nanotube and complements the AFM results. Direct tether extraction is also an important source of information about the elastic properties of the cell membrane. During the nanotube extrusion, force curves have been acquired and nanotube elasticity has been calculated. A typical force-deformation curve in tether extraction is shown in Figure 8B.

Figure 7B shows the comparison between the three conditions. There is an initial situation with the trapped microsphere attached to the cell and then a clear change in the slope when cell membrane deformation and initial tether formation begins. This part displays a monotonically rising portion and it slope provides the nanotube extrusion resistance. Fibro-CT and WT had a stretch resistance mean value of 9.7 ± 3.31 pN/µm, while in the mutated condition, nanotubes present a higher resistance mean value of 11.4 ± 4.71 pN/µm.

## 3. Discussion

### 3.1. Lamin A/C D192G Gene Mutation Effect on the LINC Complex

Previous studies described the role of the LINC complex on anchorage of cytoskeleton in nuclear membrane allowing its stability [32,41,42,43]. The LINC complex is composed of outer and inner nuclear membrane Klarsicht, ANC-1, and Syne homology (KASH), and Sad1 and UNC-84 (SUN) proteins. LINC connects the nucleus to cytoskeletal filaments and performs several functions such as nuclear positioning, mechanotransduction, meiotic chromosome movements, and cell stabilization [44,45,46,47]. In our study, immunofluorescence studies showed actin network damage and a disorganized cytoskeleton (Figure 2) and a significant decrease in the Young’s modulus in the mutated condition. As already described in the literature, the LMNA D192G mutation induces a loss of lamina cortex integrity and impact the LINC function [13,48]. In NRVFs expressing LMNA D192G mutation, the defective lamin cortex caused a decrease of the actin filament anchorage by LINC complex protein on the nuclear membrane. The cytoskeleton is anchored on the nuclear lamina cortex by a succession of LINC proteins. From inside the nuclear membrane to outside, SUN domain, KASH domain, and Nesprin proteins attach actin filaments on the lamina cortex, the plectin protein bind intermediate filament and the Kif5E protein bind microtubule on this same lamina cortex [49]. Thus, we hypothesized that in LMNA D192G NRVFs, a damaged lamina cortex could induce an alteration of the whole cell structure, and in particular the defective cytoskeleton could impact cell functions including the extracellular communication with the TNTs’ adjacent cells.

### 3.2. Impact of LMNA D192G Mutation on TNT Mechanical Properties

Actin and intermediate filaments showed disorganization of the cytoskeleton not only in the cell but also inside the TNTs in mutant-expressing fibroblasts. The literature confirms the impact on actin network disorganization in cells expressing LMNA D192G mutation [7,13], but these studies were carried out on cardiomyocytes. Our results demonstrate that the mutation D192G is also expressed in other cardiac cells like cardiac fibroblast and influences their overall cytoskeleton organization including the TNT structure. TNTs were characterized by a length between 50 to 270 nm, with cytoskeleton filament inside [19,23] (Figure 2). In the mutated NRVFs, actin filaments inside the TNT (in red) were reduced in comparison with intermediate filaments. This decrease could weaken the TNT structure. Furthermore, the information exchange by TNT between two cells could be altered and impact the fibroblasts functions in the cardiac tissue. Indeed, AFM tests confirmed that in Fibro-MT, both elasticity (cells are more deformable) and adhesion properties (Figure 3) were reduced when compared to Fibro-WT. Furthermore, AFM data proved that the separation between two Fibro-MT requires less force than the force required to separate two Fibro-WT. The analysis of the AFM retracting curves [35,39,50] also provided the distribution on the nanotube breaking force, highlighting the presence of several force populations (Figure 6). A model from Dérényl et al. [50] proposes that each force population represents TNTs with similar breaking points but in different numbers (the first population represents the breaking force of a single nanotube, the second population the breaking force of two nanotubes, etc.). A second model by Sun et al. [51] proposes that each population is considered a breaking force of specific nanotube family. These breaking forces are impacted by the sum of all macromolecular contributions to nanotube formation. In this model, the type of nanotube is dependent on the cytoskeleton properties and interaction of adhesion proteins during the contact step. Our results offered a better fit with the second model, in which one nanotube population extruded from the mutated condition has been impacted by disorganization in the cytoskeleton and by altering the function of one adhesion protein such as Cx43, which it is necessary for priming TNT formation. As previously mentioned, in the WT condition, one nanotube population has been detected with a mean breaking force around 60 pN. This population has no longer been detected after Cx43 blocking by Gap 27 (Figure 6E). In the mutated condition, this population with a breaking force around 60 pN decreases significantly. Therefore, on the membrane surface of NRVFs expressing the LMNA mutation, the nanotubes’ formation could be impacted by alteration on the Cx43 protein, as observed in WT condition after blocking Gap 27. AFM experiments allowed us to quantify the adhesion force during the cell-cell separation and the nanotube formation. In the LMNA mutant condition, these parameters were altered: adhesion properties were weaker, and the generated TNTs presented lower breaking force values.

### 3.3. Extrusion of Single Nanotube by OT

As shown in Figure 7, the elasticity of nanotube extrusion by OT bead in the NRVF-MT is 17.5% higher than in the NRVF-WT condition. To understand this trend, a comparison between the AFM and OT test parameters must be considered: in particular, the size of contact surface between bead and cell, interaction types in the contact zone and the force applied during the contact time. During OT experiments, TNTs have been extruded after a contact between the bead in laser trap and the membrane surface. The size of this contact zone is lower that the bead diameter, therefore lower than 1 µm², while during AFM acquisition, the size of contact surface was equal to the NRVF size immobilized on the cantilever surface (around 100 µm²). Thus, cell mechanics (resistance to mechanical stress, change in cell shape, membrane mechanical parameters) involved during OT and AFM tests are quite different. Moreover, the force applied during OT test is about 40 pN vs. 2 nN in the AFM cell–cell adhesion measurements. Therefore, the force applied in the OT experiment involves a quite different cell mechanical response than during the AFM measurements [52,53]. As described by H. Moysés Nussenzveig [54], the nanotube extruded using OT consists of a bilayer plasma membrane and is different from nanotubes studied in AFM tests. In particular, three membrane phenomena influence the extrusion of the bilayer membrane nanotube: cell membrane bending rigidity [55], surface tension [50,56,57], and membrane reservoir [58]. As shown in Figure 7C,D, the bending rigidity of the bilayer membrane arises from the fact that during TNT extraction the lower membrane layer is compressed and the upper one is stretched. This stress affects the bilayer equilibrium spacing, and it requires energy input to extrude the membrane nanotube. A second parameter is the surface tension, this resistance of nanotube extrusion results from the difference in pressure across the bilayer. The pressure differences arise from the pulling force by the OT bead and from cytoskeleton filaments bounded to the membrane (Figure 7C). The last phenomenon represents a fundamental mechanism of cell shape change: reservoir membrane. As described in the literature [58], cells have a plasma membrane unfolding to keep their capacity to use membrane extension for different biological mechanisms such as growth [59] and cytokinesis [60]. However, even considering these parameters, the nanotube extrusion in the mutated condition has higher elasticity than in the WT condition. Indeed, to extrude a nanotube in the WT condition, the force applied is higher to the bending rigidity and surface tension in order to unfold the reservoir membrane. In the mutated condition, the cytoskeleton damaged induces a decrease of surface tension. Therefore, the force applied to have access to the reservoir membrane and extrude a nanotube should be lower in the mutated condition than in the WT condition. However, the study of L. Figard et al. [58] characterized the reservoir membrane mechanism and demonstrated that the stabilization of the reservoir membrane is performed by the actin filament. In the NRVF expressing LMNA D192G mutation, the actin network is damaged and induces an alteration on the reservoir membrane function. In the mutated condition, this network disorganization decreases the reservoir membrane access during the nanotube extrusion experiments performed by OT. In Figure 7B, the absence of significant differences between mutated and WT conditions could be explained by the equilibrium between the reservoir membrane alteration and the higher cell deformability induced by cytoskeleton damage. Moreover, cell size has been measured (Appendix A) to determine if the reservoir membrane alteration has an impact on the cell size. A significant difference in measurements of cell size have been observed. In the mutated condition, NRVFs present both less length and width than CT and WT condition. These results could explain a decrease on the deformability capacity in mutated condition when compared with the NRFVs in WT condition.

### 3.4. Impact on Adhesion Protein in Mutated Condition

LMNA mutations alter the lamina cortex and induce cytoskeleton disorganization by loss of LINC complex anchorage on the nuclear membrane [48]. Without this anchor, the cytoskeleton is damaged and the capacity to generate a TNT is compromised. Girao’s works [25] showed that Cx43 is an important actor of intercellular communication, mediated by extracellular vesicles, gap junctions, and TNTs. The analysis of adhesive pattern by cell–cell adhesion shows the impacts of Cx43 (Figure 5 and Figure 6). When Cx43 proteins are blocked by Gap 27 mimetic peptide in the WT NRVFs, a decrease in the adhesion protein and nanotube population values takes place. The same population of adhesion protein decreases in the NRVFs expressing the LMNA 192G mutation (Figure 5D). This decrease suggests an alteration on the Cx43 function. The Cx43 migration to the membrane surface requires an organized cytoskeleton as suggest by Thomas et al. [61]. The transport of Cx43 to the membrane surface is dependent upon intact cytoskeletal filaments. In LMNA mutant NRVFs, the expression of Cx43 could be decreased on the membrane surface and thus, could induce altered adhesion properties during the separation of two cells. As seen previously, Cx43 proteins initiate TNT formation [25]: the nanotubes’ breaking force distribution (Figure 6D) suggests that the nanotubes population representing TNT were decreased in the mutated condition. Moreover, the number of retract curves analyzed and the number of events recognized from cell–cell adhesion acquisitions suggest that the mutated NRVFs capacity to generate points of interaction with another cell is reduced (Appendix A). Adhesive areas are lower on the cell wall of NRVFs expressing a laminar mutation in comparison with the WT NRVFs. The analysis of force curves events by cell–cell adhesion highlights that the LMNA D192G mutation could have an important impact on the TNTs’ formation and on the extracellular TNT communication with adjacent cells.

### 3.5. Impacts of the LMNA D192G Mutation in NRVFs on the Heart in ACM

The cardiac muscle is composed of several cell types. Cardiomyocytes account for roughly 75% of normal heart tissue volume, but they account for only 30–40% of cell numbers [62]. Among the remaining cells, fibroblasts have an important role because a component of the adult human cardiac muscle is also collagen type I and collagen type III, and for both, synthesis and turnover are mainly regulated by cardiac fibroblasts. Cardiac fibroblasts are also responsible for (physiological and pathological) ECM synthesis in the heart muscle and play an important role for the structural, mechanical, and electrical cardiac functions. Every cardiomyocyte is closely connected to a fibroblast in normal cardiac tissue. However, pathological states like ACM are frequently associated with myocardial remodeling involving fibrosis. Fibroblasts and myocytes are also interconnected in their response to mechanical stresses. The preservation of physiological levels of cardiac stiffness not only determines overall ventricular diastolic function but also ensures correct cardiomyocyte functionality. Cardiomyocytes are embedded within an ECM whom protein composition and levels of cross-linking can affect the physiological stiffness of the heart. In this regard, collagens are important players due to their ability to form fibrils that supply tissue stiffness. Therefore, it is evident that fibroblasts are a multipurpose dynamic cell that controls collagen and other ECM components affording the neighboring cells to properly function. Furthermore, the decrease of fibroblast elasticity could reduce the heart capacity to contract. Therefore, the presence of LMNA mutant fibroblasts necessarily impacts the overall heart function. Furthermore, ACM is a pathological condition generated by a large spectrum of genetic variations that can target the *LMNA* gene. According to our results, the mutation of the lamina cortex could induce the decrease of extracellular communication by TNT during ACM pathology. Lack of communication from fibroblasts in cardiac tissue could bring fibrofatty infiltration in cardiomyocytes with tissue fibrosis [63,64,65]. Several reports illustrated the function of fibroblasts in the ECM production and their role in fibrosis and adipogenesis [63,64,65,66]. Cells in the cardiac tissue secrete fibrogenic mediators and matricellular proteins that bind to the fibroblast surface receptors. Then, the fibroblast receptors transduce intracellular signaling cascades that regulate genes involved in synthesis, processing, and metabolism of the extracellular matrix. Fibroblast endogenous pathways are involved in the negative regulation of fibrosis and are critical for cardiac tissue as they may protect the myocardium from excessive fibrogenic responses. Therefore, cellular communication is necessary to receive and exchange the biological information in order to have an adapted regulation response. In ACM cells, LMNA mutations can modify this regulation by a communication of altered TNTs. The restoration of TNT formation in cardiac cells of ACM patients could be an interesting research perspective to decrease cardiac damages.

### 3.6. Study Limitations

A limitation of our study is that an important assumption has been made, specifically that the cohesive force between cells were only due to the nanotubes, while the junctions formed by desmosomal proteins, adherens junctions, or CAMs have not been taken into account. In our study, the cells confluence was relatively low and therefore TNTs were the principal players of the adhesion between cells.

Furthermore, although the study of pro-fibrotic and pro-adipogenic changes in LMNA mutant cardiac fibroblasts was beyond the scope of our study, it should be noted that Chen et al. have previously reported altered mechanotransduction/mechanosignalling in LMNA mutant cardiomyocytes, leading to the activation of the Hippo pathway, that triggers fibro-adipogenic signals in fibroblasts [67]. Finally, other mechanisms may be implicated in the defective mechanotransduction of LMNA mutant cardiac cells. In cardiomyocytes with three different LMNA missense mutations, we previously reported defective adhesion, altered actin, and microtubule networks and dislocated connexin 43 [10] leading to higher frequency of beating but with a reduced beating force. However, future studies should investigate the complex molecular changes in LMNA mutant cardiac fibroblasts.

## 4. Materials and Methods

### 4.1. Isolation and Culture of Ventricular Fibroblasts from Neonatal Rat (NRVFs)

The University of Colorado Denver institutional review board gave approval for these experiments (protocol number 00235-30 July 2019). All animal studies have been carried out according to the guidelines Animal Care and Use Committee. All animal experiments were performed using all possible methods to alleviate or minimize potential pain, suffering, or distress, and enhance animal welfare. Animals were provided with housing in an enriched environment, with at least some freedom of movement, food, water, and daily care and cleaning. The well-being and state of health of experimental animals were observed by competent persons, dedicated to the management of the Animal House, able to prevent pain or avoidable suffering, distress, or lasting harm. Experiments were performed solely by competent authorized persons. Primary cultures of neonatal rat ventricular fibroblasts (NRVFs) were isolated and cultured from 1 to 3-day-old Sprague Dawley rat pups (Charles River), following decapitation and enzymatic digestion as previously described with minor modifications [68,69]. Briefly, ventricles were separated from the atria using scissors and then dissociated in CBFHH buffer (calcium and bicarbonate-free Hanks with Hepes) containing 0.5 mg/mL of Collagenase type 2 (Worthington Biochemical Corporation, Lakewood, NJ, USA), and 1 mg/mL of Pancreatine (Sigma-Aldrich, St. Louis, MO, USA). To separate Myocytes and Fibroblasts, mix cells were incubated on tissue culture plates at 37 °C for 1 h. Then, the cells were washed with minimum essential media (MEM), supplemented with 5% bovine calf serum to collect the unattached myocytes, and the adhered NRVFs were further used for our experiments. After one day of culture, the NRVFs were re-plated at a density of 2 × 10^5^ cells/mL in primary Petri dishes (Falcon) [70] for further experiments.

### 4.2. Isolation Adenoviral Constructs and Infection

Methods for adenoviral infection have been previously reported [71,72]. In brief, shuttle constructs were generated in Dual CCM plasmid DNA containing GFP gene and human *LMNA* cDNA. Constructs were bicistronicity with the two inserts (*LMNA* and GFP) driven by two different CMV promoters to identify cells expressing LMNA protein using GFP as a marker of cellular infection. NRVFs were infected by adenoviruses at a multiplicity of infection (MOI) of 30 in serum free medium; 6 h post-infection, complete medium was replaced, and the cells were incubated at 37 °C and 5% CO_2_. Previous control experiments showed that GFP transfection and expression did not affect endpoints of interest in NRVF in our experimental conditions.

### 4.3. Immunofluorescence

NRVFs were fixed in PBS containing 4% PFA for 15 min at room temperature. Cells were permeabilized at room temperature, with 1% Triton X-100 for 90 min, blocked with 2% BSA in PBS for 45 min, and incubated overnight with vimentin 1:1000 (Sigma-Aldrich, St. Louis, MO, USA). Goat anti-mouse antibody conjugated to Cy5 (Abcam, Cambridge, United Kingdom) was used as secondary antibody 1:300 (Invitrogen, Waltham, MA, USA). Each sample was stained with Dapi 1:2000 (Thermo Fisher Scientific, Waltham, MA, USA) to counter-stain the nuclei and with phalloidin 1:1000 (Sigma-Aldrich, St. Louis, MO, USA) prepared in 1% BSA + 0.3% Triton 1X to staining the cytoskeleton. Representative immunofluorescence images were acquired using a Zeiss LSM780 confocal.

### 4.4. Single Cell Force Spectroscopy by AFM

Atomic force microscopy is widely used for measuring the elasticity (Young’s modulus) of living cells. In this case, AFM experiments were carried out by single cell force spectroscopy on NRVFs with complete medium, at 37 °C and on immobilized NRVFs using a human fibronectin coating (20 µg/mL). During acquisition, MLCT AFM probes were used. These probes have a pyramidal tip made of Si_3_N_4_, with a curvature radius of 35° and are manufactured by Bruker. The cantilever spring constants were systematically checked using the thermal tune method and were found to range from 10 to 30 pN/nm [73]. The maximal force applied to the cell was limited to 2 nN to preserve the cell membrane integrity. The acquired force (F) versus displacement curves were converted into indentation (δ) curves and fitted with the Hertz–Sneddon model [74]. The AFM experiments were carried out at a velocity of 5 µm/s [75,76]. All experiments were performed at the same velocity since AFM tests are rate-dependent.

### 4.5. Cell–Cell Adhesion Experiments by AFM

#### 4.5.1. Cell Preparation

For AFM cell–cell adhesion assay [36], two areas were demarcated in the TPP Petri dish: the first one was aimed at obtaining individual adherent cells on fibronectin coating after 24 h, and the second one to prepare non-adherent cells on the dish surface without coating. Briefly, the previous day, the first area was coated for 1 h at 37 °C with 20 µg/mL human fibronectin in phosphate buffered saline (PBS, Thermo Fisher Scientific, Waltham, MA, USA) and then washed twice with PBS. Then 15,000 cells were seeded on the coating side and cultured in complete DMEM medium (Thermo Fisher Scientific, Waltham, MA, USA) at 37 °C in 5% CO_2_. The day of AFM measurements, a second Petri dish with the same seeded cells were trypsinized and washed to obtain a tube of non-adherent cells. These cells were stocked at 37 °C in 5% CO_2_. Just before the AFM experiments, the non-adherent cells were added in the first Petri dish on the non-coating side at 37 °C [35].

#### 4.5.2. Data Acquisition

The cell biomechanical assessments were performed with the CellHesion 200 apparatus (JPK Instruments Bruker Nano GmbH, Berlin, Germany), which provides a large vertical piezoelectric range of 100 µm, mounted on a Zeiss inverted optical microscope with a controlled temperature of 37 °C. Tipless cantilevers from Bruker (Bruker, Billerica, MA, USA) (NP-O10) were used with the nominal spring constant of 30 mN/m, the exact value for each cantilever was inferred from the thermal tuning method [72]. For each experimental condition, the cantilever was replaced after every 3 cell contacts. The cantilevers were coated with fibronectin (10 µg/mL for 30 min before acquisition) after being activated by ozone (0.1 mBar, 3 min) in order to promote cell attachment. The cell-adherent cantilever was brought into contact with one non-adherent cell localized in the zone without fibronectin coating on the Petri dish (providing low adhesion to the substrate) (Figure 4A). This “fishing” step was carried out with a contact duration of 30 s applying 1.5 nN. After 30 s the cantilever was retracted and stopped in this position for 1 min to allow the stabilization of cell caught at the end of the cantilever. The approach–retraction steps were performed with a ramp size of 80 µm, cantilever speed of 5 µm/s, contact time of 25 s, and force setpoint of 2 nN.

#### 4.5.3. Data Analysis

The acquired force curves were analyzed using the JPK Data processing software. The first numerical treatment was the reduction of curve noise, followed by the baseline definition and the tilt correction. After these corrections, several adhesion parameters, identifiable on the retract force curve, were measured, exported, and analyzed (Figure 4B): in particular (i) the total energy applied during the separation of 2 cells, represented by the area under the retracting force distance curve; (ii) the maximum force is the maximum value of the force applied during the separation step; (iii) the rupture pattern which illustrates interaction during short distance separation (0 to 5 µm of piezo withdraw). This pattern is characterized by jumps in the force retracting curve everyone preceded by a slope >20%. These ruptures events were induced by broken adhesion protein link between two cells [36,50]; (iv) nanotube pattern i.e., nanotubes generated during long-distance separation (5 to 80 µm of piezo withdraw). In this latter case, this pattern of jumps in the retract force curve was preceded by a plateau with <15% derivation and persisting for > 1 μm, representing the force needed to break these plasma membranes, pulled out to form a nanotube between two cells after their contact [35,50,51,77]. The distribution of breaking forces was analyzed using OriginPro 9.7 2020 software, and Gaussian function was fitted to highlight a specific population.

#### 4.5.4. Synthetic Connexin 43 Mimetic Peptide (Gap 27) for Cx43 Blocking

In this study, Gap 27 was used to block the function of Cx43 gap junction. Gap 27 is mimicking amino acids 204–214 on extracellular loop 2 of Cx43 (SRPTEKTIFII). Before the AFM tests of cell–cell adhesion, selected cell cultures were incubated with Gap 27 (Sigma-Aldrich) in PBS at 150 µM for 1h, at 37 °C [78,79,80].

### 4.6. Optical Tweezers

In order to extract the nanotubes’ membrane and measure their stiffness, a custom optical tweezers was used with the technique previously described [81] and summarized in Figure 8A.

An infrared trapping laser from an Ytterbium fiber laser (YLM-5–1064-LP, IPG Photonics, Oxford, MA, USA) with a wavelength of 1064 nm and maximum power of 5 W at a nominal power of 250 mW (60 mW at the sample) was used. The vertical stiffness of the optical trap was k_z_ = 0.04 pN/nm. A silica microbead of 1 µm diameter (SS04000 Bangs Lab Inc., Fishers, IN, USA) was used to extract the tether membrane. This step was achieved by manipulating the bead to get in contact with the cell, then moving the cell/stage vertically against the bead by 0.5 µm and keeping the contact for 30 s to create the cell adhesion. Finally, a nanotube was extracted by moving the cell away from the bead by 2 µm. Precise displacement of the bead was obtained by moving the optical trap with a Focus Tunable Lens FTL (EL-10-30-C, Optotune AG, Dietikon, Switzerland), as previously described [82]. The displacement of the cell, which was adhered on the coverslip, was controlled in x-y-z by a 3-Axis NanoMax Flexure Stage (MAX311D/M Thorlabs Inc., Newton, NJ, USA). The displacement of the bead from the center of the trap, which allows us to measure the force, was measured by a Quadrant Detector Sensor (PDQ080A, Thorlabs Inc., Newton, NJ, USA). The vertical force exerted by the tether during extraction was calculated as F = k_z_B and the length of the tether L was given by the displacement of the nanopiezo. The nanotube’s stiffness was deduced as the slope of the force–tether length (F–L) curve (Figure 8B).

### 4.7. Statistics

For all experiments, data was collected from at least three independent experiments. All data were first subjected to the Shapiro–Wilk normality test. The unpaired one-way ANOVA with Dunnet’s or Tukey’s correction for normal distributions or the Kruskal–Wallis with Dunn’s correction test was employed. A confidence interval of 95% (*p* value of <0.05) was used to identify significant differences among the samples. Data in the text are reported as mean of values ± standard deviation and graphically presented as scattered dot plots with mean (and median for those deviating from normality test) and standard deviation.

Statistical analysis was performed using GraphPad Prism 7 software (San Diego, CA, USA).

## 5. Conclusions

The goal of this study was to define the multiple impacts of the mutation LMNA D192G on cardiac fibroblast cellular mechanisms, such as cytoskeleton anchorage on nuclear membrane, cell deformability, cell shape evolution, adhesive properties, and TNT formation. Using a multidisciplinary approach including immunofluorescence, AFM, and OT, we investigated the changes in biomechanical properties and cell–cell adhesion of cardiac fibroblasts with the LMNA D192G mutation, which can cause an inherited arrhythmogenic cardiomyopathy. We found that mutant NRVFs have a damaged cytoskeleton, as previously shown in cardiomyocytes [7], leading to a decrease of cell elasticity and a lower force of adhesion between two NRVFs. Furthermore, we found altered mechanical properties of nanotubes generated by contact with another cell (by AFM) or by contact with a silicate bead (using OT). Our results complement our previous investigations on LMNA mutant cardiomyocytes [7,67] and suggest that this mutation also impacts biomechanical properties of cardiac fibroblasts. Future studies should investigate the impact of the LMNA mutations on TNT communication in each type of cardiac cells and also the TNT communication at a higher scale, such as in the cardiac tissue.

## Figures and Tables

**Figure 1 ijms-22-09193-f001:**
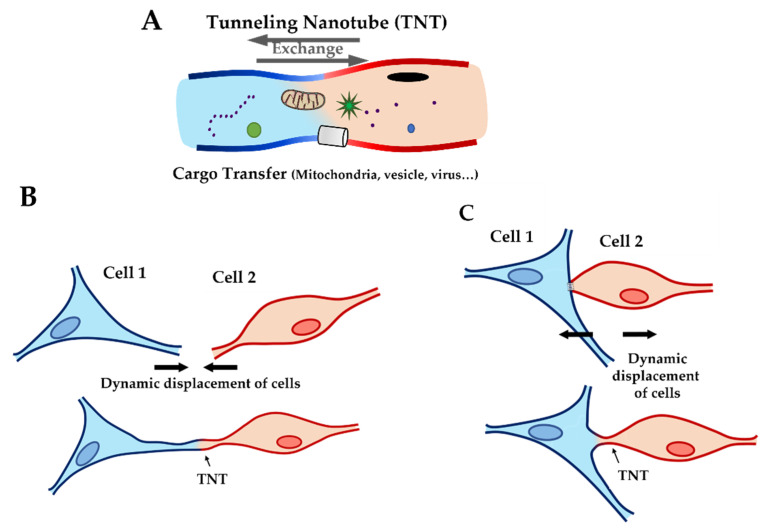
Cartoon showing tunneling nanotubes (TNT). (**A**) Cargo transfer through TNT with exchange of particles (such as mitochondria, ions, virus…) between two cells. (**B**) TNT construction by extension of filopodia. (**C**) TNT construction by contact, formation of TNT after physical contact between two cells and distancing of cells.

**Figure 2 ijms-22-09193-f002:**
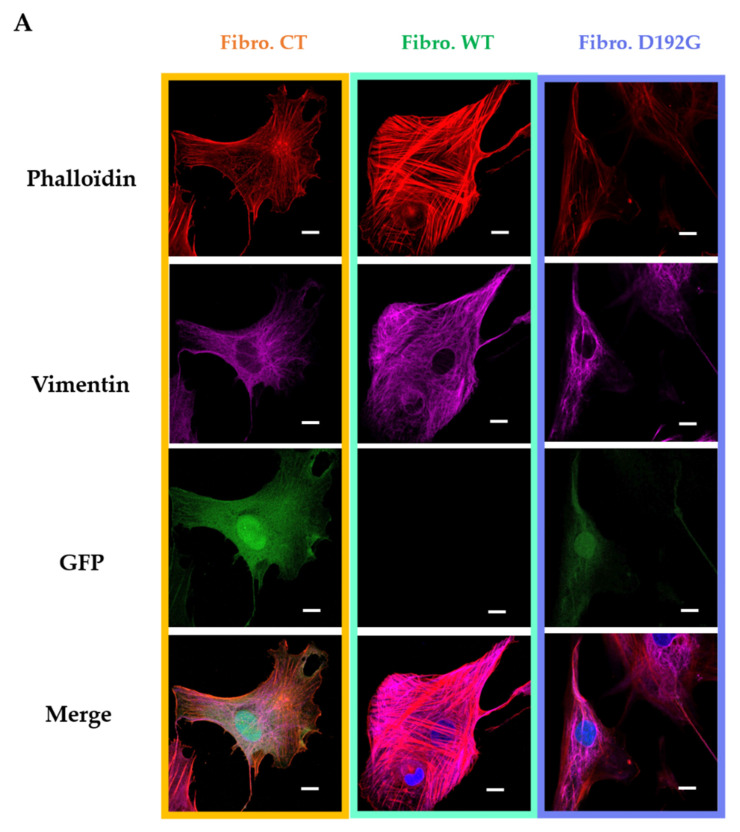
(**A**) Fluorescence staining of NRVF cytoskeleton. NRVF fluorescence staining in CT, WT, and lamin A/C D192G mutated condition with (i) Phalloidin to target both actin filaments and vimentin intermediate filaments, (ii) DAPI to target the nucleus, and (iii) green fluorescent protein (GFP) used as the transfection control after infection of adenoviral with, or without, the mutation lamin A/C D192 G. Scale bar (white), 10 µm. (**B**) Fluorescence staining of NRVF nanotubes. NRVF nanotubes’ fluorescence staining in CT, WT, and lamin A/C D192G mutated condition with (i) Phalloidin to target both actin filaments and vimentin intermediate filaments, (ii) DAPI to target the nucleus, and (iii) green fluorescent protein (GFP) used as the transfection control after infection of adenoviral with, or without, the mutation lamin A/C D192 G. Scale bar (white), 10 µm.

**Figure 3 ijms-22-09193-f003:**
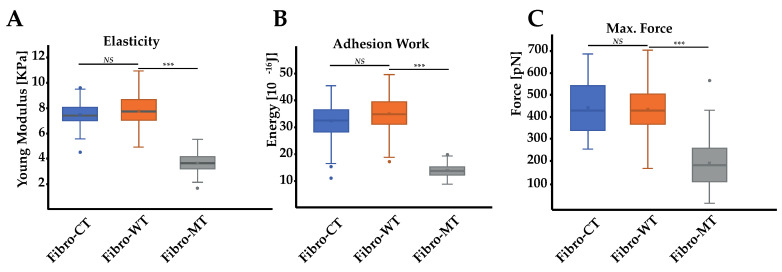
Biophysical properties of living NRVFs. (**A**) Comparison of elasticity (expressed as Young’s modulus) between NRVFs with the WT adenoviral infection (Fibro-CT), NRVFs wild type (Fibro-WT), and NRVFs with *LMNA* D192G mutation (Fibro-MT) (n^Fibro-CT^ = 61, n^Fibro-WT^ = 63, n^Fibro-MT^ = 56 analyzed curves, respectively). (**B**) Cell–cell adhesion energy defined as the energy to separate two NRVFs. (n^Fibro-CT^ = 152, n^Fibro-WT^ = 221, n^Fibro-MT^ = 177 analyzed curves, respectively). (**C**) Comparison of maximum force in cell–cell adhesion/de-adhesion expresses as the maximum force applied to separate two NRVFs, (n^Fibro-CT^ = 152, n^Fibro-WT^ = 221, n^Fibro-MT^ = 177 analyzed curves, respectively), (*** *p* < 0.0001, *NS* = non-significant). Numerical data is divided into quartiles, and a box is drawn between the first and third quartiles, with an additional line drawn along the second quartile to mark the median. The minimums and maximums outside the first and third quartiles are depicted with lines.

**Figure 4 ijms-22-09193-f004:**
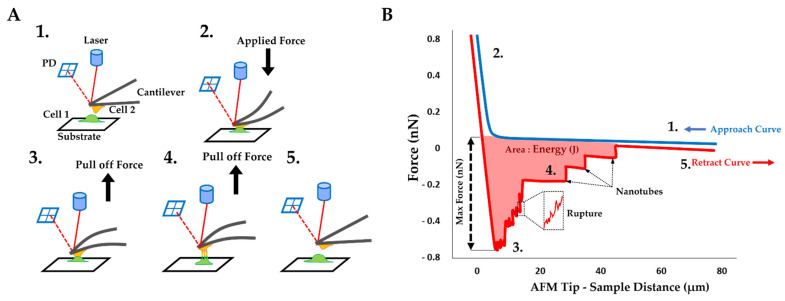
Cell-cell adhesion experiments by AFM. (**A**) Cell-cell adhesion setup: 1: Initial position with a fibroblast (cell 1) immobilized by fibronectin on support and a second cell (cell 2, yellow) attached to a tipless AFM cantilever coated with fibronectin driven by a wide-range (100 μm) piezo element. 2: Contact between cells by vertical force applied on the cantilever. The force on the cantilever is monitored by the deflection of a laser beam focused on the cantilever end and recorded by the photodiode (PD). 3: Cantilever retract step, producing a cells separation by a short distance. 4: Cantilever retract step, separation of cells over a longer distance and TNT formation. 5: Return to the initial step and total separation between cell 1 and cell 2. (**B**) Force–curve principle. Characteristic force curve recorded by PD, representing the force on the cantilever and the displacement during a cell–cell detachment and separation experiment: during the approach step (blue), the cell on the cantilever is brought into contact at a constant velocity, then the cell is retracted (red curve), until it goes back to the initial position going through a different pattern.

**Figure 5 ijms-22-09193-f005:**
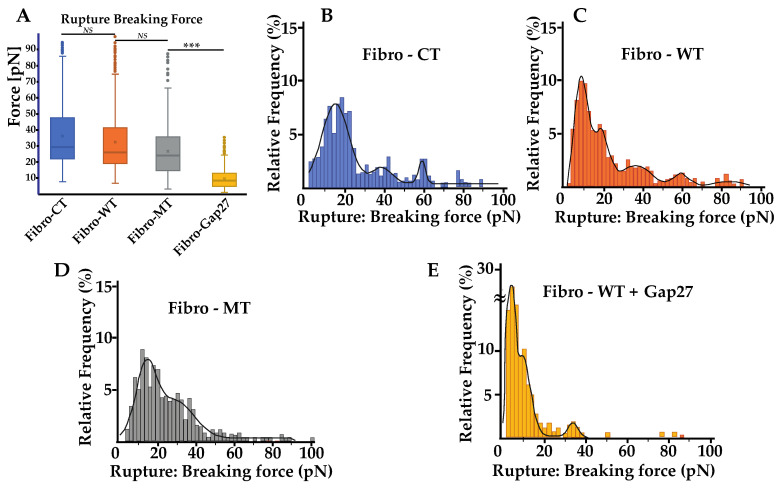
Rupture pattern obtain using the AFM. (**A**) Comparison of rupture pattern between control, wild type, *LMNA* D192G mutation, and NRVFs WT incubate with Gap27 (Fibro-WT + Gap 27). Rupture breaking force is the force applied to break the link between the adhesion proteins of two cells in contact during the “short distance (0 to 5 µm)” de-adhesion step. (n^Fibro-CT^ = 152, n^Fibro-WT^ = 221, n^Fibro-MT^ = 177, n^Fibro-WT + Gap27^ = 122 analyzed curves, respectively). (**B**–**E**) Rupture breaking force distribution histogram for the different NRVFs conditions. The black curves represent the overall Gaussian fit distributions, (*** *p* < 0.0001; *NS* = non-significant).

**Figure 6 ijms-22-09193-f006:**
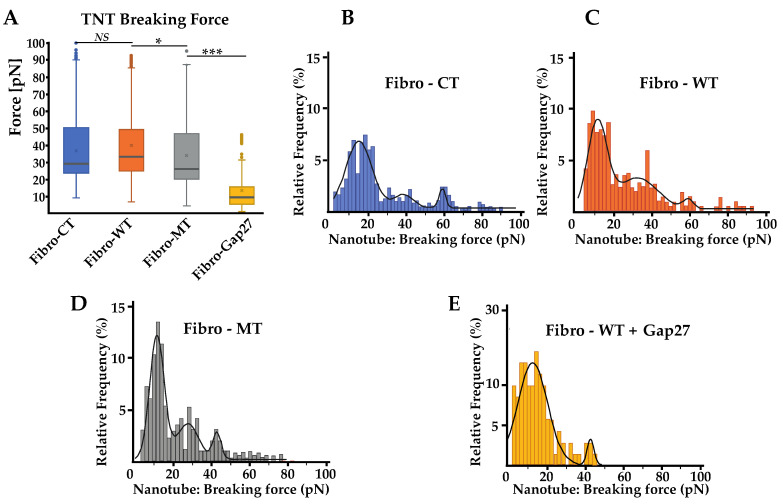
Nanotube pattern obtain using the AFM. (**A**) Comparison of nanotube pattern between NRVF control and wild type, NRVFs with *LMNA* D192G mutation and NRVFs WT incubate with Gap 27. Nanotube breaking force represents the force applied to break the link between both cells during the “long distance (5 to 80 µm) de-adhesion steps”. (n^Fibro-CT^ = 152, n^Fibro-WT^ = 221, n^Fibro-MT^ = 177, n^Fibro-WT + Gap27^ = 122 analyzed curves, respectively). (**B**–**E**) Breaking force distribution histogram of nanotube pattern in different NRVF conditions. The black curves represent the overall Gaussian fit distributions. (*** *p* < 0.0001; * *p* < 0.05, *NS* = non-significant).

**Figure 7 ijms-22-09193-f007:**
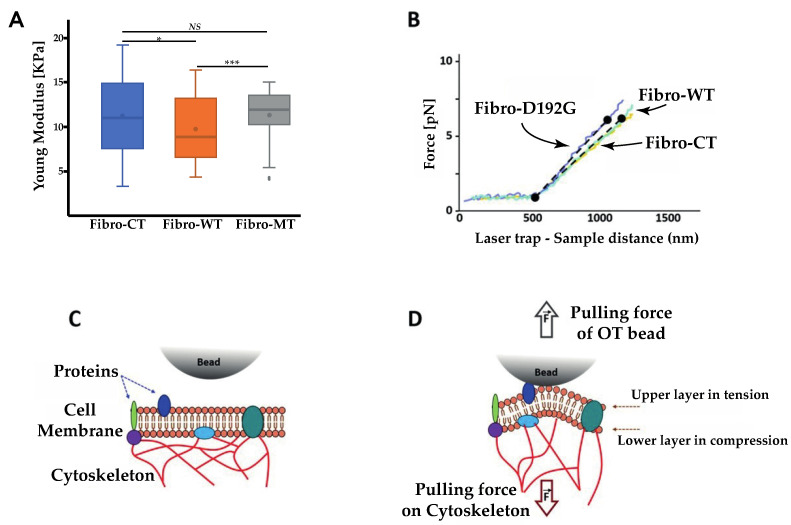
Nanotube extrusion measured using the optical tweezers. (**A**) Nanotube stiffness (expresses as Young’s modulus) between different conditions NRVFs. (n^Fibro-CT^ = 60, n^Fibro-WT^ = 58, n^Fibro-MT^ = 70, analyzed curves, respectively). (**B**) Force curve between NRVFs CT, WT, and NRVFs with *LMNA* D192G mutation. (**C**) Cartoon showing the resistance force impacting the nanotube extrusion. (**D**) The pulling force of the OT bead after the interaction with the plasma membrane is counteracted by bending rigidity (with stretching and compression of membrane layer) and surface tension (with pulling force of cytoskeleton), (*** *p* < 0.0001; * *p* < 0.05; *NS* = non-significant).

**Figure 8 ijms-22-09193-f008:**
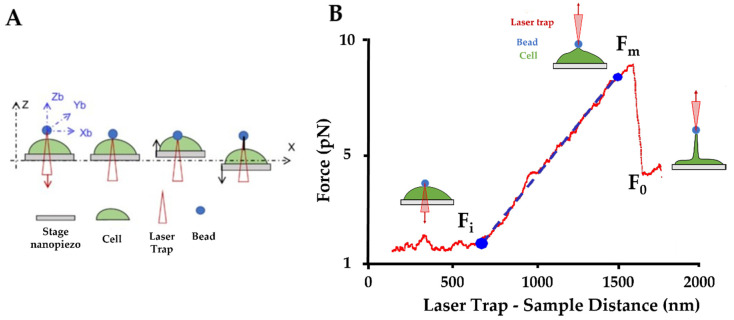
Optical tweezers (OT) setup. (**A**) Schematic procedure of an optically trapped bead applying a force to form a nanotube from the cell membrane. The trapped bead can be moved in three directions (Z_b_/Y_b_/X_b_), in this way the bead can be put in contact with the cell. Then the trap remains in the same position, and the stage moves in Z and X direction to extract a nanotube (black line). (**B**) Force curve obtained by OT: The initial force (F_i_) is the force, between bead and cell, measured during the contact step. The maximum force (F_m_), is the maximum force applied to extrude the nanotube from the cell surface. The plateau force (F_0_), is the constant force applied during the nanotube extension. The slope between the two points F_i_ and F_0_ is used to calculate the nanotube’s stiffness (blue line: slope used to stiffness; red curve: force applied on bead during the laser displacement).

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
