# Peer review of "Compromised Biomechanical Properties, Cell–Cell Adhesion and Nanotubes Communication in Cardiac Fibroblasts Carrying the Lamin A/C D192G Mutation"

_ijms, 2021, doi:10.3390/ijms22179193_

Round 1
Reviewer 1 Report
In the present article, the authors ‘goal was to investigate the impact of the LMNA D192G mutation on neonatal rat ventricular fibroblasts (NRVF) at the biophysical and biomechanical levels. An important originality of this paper is that the authors addressed the effect of the LMNA mutation in non-muscle cells that represent the majority of the cells present in the heart. The methods used are adapted to this goal and well-detailed throughout the manuscript, and the author’s conclusions are supported by the data provided.
I have some suggestions for improving this manuscript, notably regarding the Discussion section.
Major points
- On page 17, the authors extrapolate the breaking force necessary for separating integrin and fibronectin (that was already measured) to the hypothetic value for 2 integrins molecules (30 pN peak). I do not think the authors can state that it is “likely” the right peak as the force of interaction can be very different depending on the partner involved. It is just an hypothesis that will need confirmation as some sub-peaks may be hidden. The text should be modulated accordingly.
- There are several discrepancies between the text and the figures and the nomenclature is not consistent throughout the article. For example, p15: the maximum force values indicated in the text do not correspond to the ones that can be estimated in the Figure. Please correct the values and be consistent with the nomenclature in the text and the figure legends. It seems also to me that the groups discussed in the text are not the same than the ones indicated in Fig. 5 (Gap27 group is not present in the text). The same confusion (values and text) is also present for Fig. 6 and the text on p.15. Please correct the text accordingly.
- The Discussion section is too long with some redundancy with the result section, and by contrast may be improved by adding some important aspects. Second the authors, what is/could be the impact of cardiac fibroblasts in terms of elasticity and mechanical properties when considering the whole heart? Are they proportional to their abundance when compared to the cardiac cells? What can be their role during the development of the pathology in patients? The authors do not make the parallel between the modifications already published in cardiac cells and the work presented here.
- In the Discussion section, the authors stated that: “The LMNA D192G mutation induces a loss of lamina cortex integrity and impact the LINC function”. Was it already published? No reference is cited. If not, the assumption is highly speculative and should be proposed as an hypothesis.
Minor points
- The figures provided about the methodologies used are highly helpful for non-specialists but Figures 2 and 3 are too small and of poor resolution. Please modify accordingly
- Some typos are present throughout the text. See for example p.12, 4th paragraph: “ As far as the TNT mechanical properties are concerned …”; p13: One should read, Suppl. Figure 2 instead of Suppl. Figure 1)
- Figures 6 and 7: panel E is missing in the Figures. It seems that the Fibro-CT group is missing in the Figures, which also modifies the citation in the text (g. p.17, Fig. 6D is in fact Figure 6C, etc.). When changing the Figures and/or the text, please pay attention to cite the right panel.
Author Response
We would like to express our gratitude to the Reviewer for revising our manuscript and the advices to improve it.
- On page 17, the authors extrapolate the breaking force necessary for separating integrin and fibronectin (that was already measured) to the hypothetic value for 2 integrins molecules (30 pN peak). I do not think the authors can state that it is “likely” the right peak as the force of interaction can be very different depending on the partner involved. It is just an hypothesis that will need confirmation as some sub-peaks may be hidden. The text should be modulated accordingly.
We follow the suggestion of the Reviewer and we added at page 17 this statement “Even though some sub-peaks may be hidden, this assumption allows to compare the behavior of the different cell lines”
- There are several discrepancies between the text and the figures and the nomenclature is not consistent throughout the article. For example, p15: the maximum force values indicated in the text do not correspond to the ones that can be estimated in the Figure. Please correct the values and be consistent with the nomenclature in the text and the figure legends. It seems also to me that the groups discussed in the text are not the same than the ones indicated in Fig. 5 (Gap27 group is not present in the text). The same confusion (values and text) is also present for Fig. 6 and the text on p.15. Please correct the text accordingly.
We are sorry but for a mistake during the uploading of the paper the figures were those of a previous version with less data point. Now we have the right figures and data
- The Discussion section is too long with some redundancy with the result section, and by contrast may be improved by adding some important aspects. Second the authors, what is/could be the impact of cardiac fibroblasts in terms of elasticity and mechanical properties when considering the whole heart? Are they proportional to their abundance when compared to the cardiac cells? What can be their role during the development of the pathology in patients? The authors do not make the parallel between the modifications already published in cardiac cells and the work presented here.
Following the suggestion of the Reviewer we added this part to the discussion
The cardiac muscle is composed of several cell types. Cardiomyocytes account for roughly 75% of normal heart tissue volume, but they account for only 30– 40% of cell numbers [78]. Among the remaining cells, fibroblasts have an important role because a component of the adult human cardiac muscle is also collagen type I and collagen type III and for both, synthesis and turnover are mainly regulated by cardiac fibroblasts. Cardiac fibroblasts are also responsible for (physiological and pathological) ECM synthesis in the heart muscle and play an important role for the structural, mechanical and electrical cardiac functions. Every cardiomyocyte is closely connected to a fibroblast in normal cardiac tissue. However, pathological states like ACM are frequently associated with myocardial remodeling involving fibrosis. Fibroblasts and myocytes are also interconnected in their response to mechanical stresses. The preservation of physiological levels of cardiac stiffness not only determines overall ventricular diastolic function but also ensures correct cardiomyocyte functionality. Cardiomyocytes are embedded within an ECM whom protein composition and levels of cross-linking can affects the physiological stiffness of the heart. In this regard, collagens are important players due to their ability to form fibrils, that supply tissue stiffness. Therefore, it is evident that fibroblasts are a multipurpose dynamic cell that control collagen and other ECM components affording the neighboring cells to properly function. Furthermore, the decrease of fibroblast elasticity could reduce the heart capacity to contract. Therefore, the presence of LMNA mutant fibroblasts impacts necessarily the overall heart function. Furthermore, ACM is a pathological condition
[78] P. Camelliti, T. K. Borg, et P. Kohl, « Structural and functional characterisation of cardiac fibroblasts », Cardiovasc. Res., vol. 65, no 1, p. 40‑51, janv. 2005, doi: 10.1016/j.cardiores.2004.08.020.
- In the Discussion section, the authors stated that: “The LMNA D192G mutation induces a loss of lamina cortex integrity and impact the LINC function”. Was it already published? No reference is cited. If not, the assumption is highly speculative and should be proposed as an hypothesis.
The information was already published and we are sorry that we failed to mention it. Now we added
“As described in the literature, the LMNA D192G mutation induces a loss of lamina cortex integrity and impact the LINC function [13], [64]”.
[13] Sbaizero Orfeo et al., « Altered Nuclear and Cytoskeletal Mechanics and Defective Cell Adhesion in Cardiac Myocytes Carrying the Cardiomyopathy LMNA D192G Mutation », Circulation, vol. 130, no suppl_2, p. A17200‑A17200, nov. 2014, doi: 10.1161/circ.130.suppl_2.17200.
[64] L. Yang, M. Munck, K. Swaminathan, L. E. Kapinos, A. A. Noegel, et S. Neumann, « Mutations in LMNA Modulate the Lamin A - Nesprin-2 Interaction and Cause LINC Complex Alterations », PLoS ONE, vol. 8, no 8, août 2013, doi: 10.1371/journal.pone.0071850.
Minor points
- The figures provided about the methodologies used are highly helpful for non-specialists but Figures 2 and 3 are too small and of poor resolution. Please modify accordingly
Done
- Some typos are present throughout the text. See for example p.12, 4th paragraph: “ As far as the TNT mechanical properties are concerned …”; p13: One should read, Suppl. Figure 2instead of Suppl. Figure 1)
Done
- Figures 6 and 7: panel E is missing in the Figures. It seems that the Fibro-CT group is missing in the Figures, which also modifies the citation in the text (g. p.17, Fig. 6D is in fact Figure 6C, etc.). When changing the Figures and/or the text, please pay attention to cite the right panel.
As aforementioned, by mistake during the uploading of the paper the figures were those of a previous version with less data point (control data were missing). Now we have the right figures and data

Reviewer 2 Report
Major points:
- This variant is not described as associated to ACM (National Center for Biotechnology Information. ClinVar; [VCV000066911.3], https://www.ncbi.nlm.nih.gov/clinvar/variation/VCV000066911.3 (accessed July 5, 2021).) Therefore a clinical characterization of patients who are carriers of this variant would help to understand the expected phenotype.
- Please detail the protocol number- date of approval of the animal ethic committee approval
- Since adenovirus causes a transient transfection, how did you make sure that your cells are still expressing the episome after the experiment?
- An assumption is made that the cohesive force between cells is only due to the nanotubes, while the junctions formed by desmosomal proteins or adherens junctions or CAMs have not been taken into account. Please explain if this is true and eventually state it in the limitations. I think cell confluence is pivotal for the formation of nanotubes. Please detail.
- In Figure 6A and 7A the variability looks enormous. Please detail the number of samples used and what the error bar stand for. Probably, they need more replicates in order to be convincing.
- Page 24, last sentence, please cite updated literature on ACM fibrosis, and not only generic fibroblast contribution to fibrosis, i.e. doi: 10.3389/fphys.2020.00279; 10.3390/ijms22052673; 10.1161/CIRCRESAHA.115.308136
- Are LMNA mutant cells more pro-fibrotic? Are they more pro-adipogenic? An ACM-related phenotypic characterization is needed in order to link the mechanism (very well described) to the disease. Moreover, please discuss further (other than fibrosis) potential links between biomechanical alterations, cell-cell adhesion and nanotubes impaired communication and the disease. Indeed mechanotransduction is implicated in different phenomena in ACM.
Minor points:
- In figure 5, I cannot see (Fibro-CT) and (Fibro-MT) labels.
- The title of graph 5C is “Force Maximun”, I guess you mean “Maximum Force”
- In general, the legend lack statistical details (e.g mean+/- standard deviation?) tests used and the experimental number of replicates (either technical or biological).
- two acronims are used for arrhythmogenic cardiomyopathy: ACM and AMC. Please correct
Author Response
We would like to express our gratitude to the Reviewer for revising our manuscript and the advices to improve it.
- This variant is not described as associated to ACM (National Center for Biotechnology Information. ClinVar; [VCV000066911.3], https://www.ncbi.nlm.nih.gov/clinvar/variation/VCV000066911.3 (accessed July 5, 2021).) Therefore a clinical characterization of patients who are carriers of this variant would help to understand the expected phenotype.
Please note that ClinVar [VCV000066911.3] reports in the citations for this variant the paper by Sylvius et al. Journal of Medical genetics 2005 where the phenotype consistent with ACM is described
- Therefore, a clinical characterization of patients who are carriers of this variant would help to understand the expected phenotype
Following the Review suggestion, we added in the Introduction:
This mutation was selected because it represents a mutation associated with disruption of nuclear envelope morphology1. It has been described in one family with a severe phenotype where symptom onset and death/transplantation happened at 30.5 (±6.4) and 32.0 (±7.1) years, respectively: both patients died before the age of 40 years2. Although the small number of observations of this rare mutation must be taken into consideration, clinical data suggest that LMNA D192G have a very severe outcome.
[1] Sylvius N, Bilinska ZT, Veinot JP, Fidzianska A, Bolongo PM, Poon S, McKeown P, Davies RA, Chan KL, Tang AS, Dyack S, Grzybowski J, Ruzyllo W, McBride H, Tesson F. In vivo and in vitro examination of the functional significances of novel lamin gene mutations in heart failure patients. J Med Genet 2005;42:639–647.
[2] Bilińska ZT , Sylvius N, Grzybowski J, Fidziańska A, Michalak E, Walczak E, Walski M, Bieganowska K, Szymaniak E, Kuśmierczyk-Droszcz B, Lubiszewska B, Wagner T, Tesson F, Ruzyłło W. Dilated cardiomyopathy caused by LMNA mutations. Clinical and morphological studies. Kardiol Pol2006;64:812–819. discussion 820-811.
- Please detail the protocol number- date of approval of the animal ethic committee approval
As suggested, we added number and date of the approved protocol (00235 – 07/30/2019)
- Since adenovirus causes a transient transfection, how did you make sure that your cells are still expressing the episome after the experiment?
We added to the Results section
Since adenovirus causes a transient transfection, to control if our cells were still expressing the episome, we checked them, during and after the experiment, using GFP fluorescence.
- An assumption is made that the cohesive force between cells is only due to the nanotubes, while the junctions formed by desmosomal proteins or adherens junctions or CAMs have not been taken into account. Please explain if this is true and eventually state it in the limitations. I think cell confluence is pivotal for the formation of nanotubes. Please detail.
Following the Reviewer suggestion, we added a new part called Study limitation, in this part we explained:
A limitation of our study is that an important assumption has been made, specifically that the cohesive force between cells were only due to the nanotubes, while the junctions formed by desmosomal proteins, adherens junctions or CAMs have not been taken into account. In our study, the cells confluence was relatively low and therefore TNT were the principal players of the adhesion between cells.
- In Figure 6A and 7A the variability looks enormous. Please detail the number of samples used and what the error bar stand for. Probably, they need more replicates in order to be convincing.
We are sorry but for a mistake during the uploading of the paper the figures were those of a previous version with less data point and without data for control cells. Now we have the right figures and data. The number of curves analyzed are reported in Supplemental table 1
- Page 24, last sentence, please cite updated literature on ACM fibrosis, and not only generic fibroblast contribution to fibrosis, i.e. doi: 10.3389/fphys.2020.00279; 10.3390/ijms22052673; 10.1161/CIRCRESAHA.115.308136
Done
- Are LMNA mutant cells more pro-fibrotic? Are they more pro-adipogenic? An ACM-related phenotypic characterization is needed in order to link the mechanism (very well described) to the disease. Moreover, please discuss further (other than fibrosis) potential links between biomechanical alterations, cell-cell adhesion and nanotubes impaired communication and the disease. Indeed mechanotransduction is implicated in different phenomena in ACM.
We appreciate the comment of the Reviewer and added a paragraph to discuss this important issue (study limitations)
Although the study of pro-fibrotic and pro-adipogenic changes in LMNA mutant cardiac fibroblasts was beyond the scope of our study, it should be noted that Chen et al. have previously reported altered mechanotransduction/mechanosignalling in LMNA mutant cardiomyocytes, leading to the activation of the Hippo pathway, that triggers fibro-adipogenic signals in fibroblasts [83]. Furthermore, other mechanisms may be implicated in the defective mechanotransduction of LMNA mutant cardiac cells. In cardiomyocytes with three different LMNA missense mutations, we previously reported defective adhesion, altered actin and microtubule networks and dislocated connexin 43 [18, 84]] leading to higher frequency of beating but with a reduced beating force. However, future studies should investigate the complex molecular changes in LMNA mutant cardiac fibroblasts.
[83] S. N. Chen et al., «The hippo pathway is activated and is a causal mechanism for adipogenesis in arrhythmogenic cardiomyopathy », Circ. Res., vol. 114, p. 454–468, (2014]
[10] D. Borin et al., « Altered microtubule structure, hemichannel localization and beating activity in cardiomyocytes expressing pathologic nuclear lamin A/C », Heliyon, vol. 6, no 1, janv. 2020, doi: 10.1016/j.heliyon.2020.e03175.
[84] E. Laurini et al., « Biomechanical defects and rescue of cardiomyocytes expressing pathologic nuclear lamins », Cardiovasc. Res., vol. 114, no 6, p. 846‑857, mai 2018, doi: 10.1093/cvr/cvy040.
Minor points:
- In figure 5, I cannot see (Fibro-CT) and (Fibro-MT) labels.
Done
- The title of graph 5C is “Force Maximun”, I guess you mean “Maximum Force”
Done
- In general, the legend lack statistical details (e.g mean+/- standard deviation?) tests used and the experimental number of replicates (either technical or biological).
Done
- two acronims are used for arrhythmogenic cardiomyopathy: ACM and AMC. Please correct
Done

Round 2
Reviewer 2 Report
With the rebuttal, the authors replied properly to some of the issues raised, while other remain major issues:
1- We are still not convinced that the mutation is associated to the ACM phenotypes. All the cited papers rather describe the involvement of this mutation in DCM. Please revise the literature and either provide evidence of ACM phenotypes in the described mutation carriers or restructure the paper in view of a DCM phenotype.
2- Statistics is not properly described both in the Methods and in the legends.
- each legend has to report the n of replicates and if they are technical or biological.
- which Anova test has been used? one or two way? where is the result of the Anova? which post-test has been used for inter-sample comparisons?
- Has the distribution normality been tested (since all the the graphs were changed to box plot)?
- What is represented by the box plot? mean, median, which error?
Author Response
- We are still not convinced that the mutation [D192G] is associated to the ACM phenotypes. All the cited papers rather describe the involvement of this mutation in DCM. Please revise the literature and either provide evidence of ACM phenotypes in the described mutation carriers or restructure the paper in view of a DCM phenotype.
The reviewer is correct that the specific D192G mutation, that was originally described by Sylvius, Tesson et al. (J Med Genet 2005), was reported in a young patient who presented with conduction disease and DCM, and the cause of cardiac death (age 27), was not specified (presumably either HF or SCD). Please note that the LMNA D192G mutation was chosen in our studies because it was reported as a mutation dramatically changing the cardiomyocyte structure (Sylvius 2005, Lanzicher 2015, Lanzicher 2015, Laurini 2018, Borin 2020).
However, in the current literature and by the most updated guidelines of international societies, it should be noted that LMNA mutations are consider one of the causes of ACM, which now by definition includes a wide range of cardiomyopathies characterized by arrhythmias and either right ventricular (ARVC), biventricular, or left ventricular (“arrhythmogenic” DCM or ALVC) involvement. Please refer to the 2019 HRS Expert Consensus Statement on Evaluation, Risk Stratification, and Management of Arrhythmogenic Cardiomyopathy (Towbin, McKenna et al., Heart Rhythm 2019; Heart Rhythm. 2019 Nov;16(11):e301-e372).
To clarify this important point, we have added a sentence in the introduction. Now the initial part of the introduction is:
Arrhythmogenic cardiomyopathy (ACM) is a myocardial disease characterized by a high risk of life-threatening ventricular arrhythmias, sudden cardiac death, and progression towards heart failure [1],[2]. This disease, which can affect predominantly the right ventricle (arrhythmogenic right ventricular cardiomyopathy, or ARVC), the left ventricle (arrhythmogenic left ventricular cardiomyopathy (ALVC), or arrhythmogenic dilated cardiomyopathy (DCM)), or both (biventricular ACM) is generated by a large spectrum of genes including the lamin A/C gene (LMNA) [1]. Many ACM genes, such as LMNA [3], induce important alterations of the signal transduction and the intercellular communication mechanism in cardiac cells [4]–[6]........
- Statistics is not properly described both in the Methods and in the legends.
each legend has to report the n of replicates and if they are technical or biological. Which Anova test has been used? one or two way? where is the result of the Anova? which post-test has been used for inter-sample comparisons? Has the distribution normality been tested (since all the the graphs were changed to box plot)? What is represented by the box plot? mean, median, which error?
Now the Materials and Methods section has this part
2.7. Statistics
For all experiments, data was collected from at least three independent experiments. All data were first subjected to the Shapiro–Wilk normality test. The unpaired one-way ANOVA with Dunnet’s or Tukey’s correction for normal distributions or the Kruskal–Wallis with Dunn’s correction test was employed. A confidence interval of 95% (p value of <0.05) was used to identify significant differences among the samples. Data in the text are reported as mean of values ± standard deviation and graphically presented as scattered dot plots with mean (and median for those deviating from normality test) and standard deviation
Statistical analysis was performed using GraphPad Prism 7 software (San Diego, CA, USA).
Every figure legend now is like this
Figure 5. Biophysical properties of living NRVFs. A) Comparison of elasticity (express as Young Modulus) between NRVFs with the WT adenoviral infection (Fibro-CT), NRVFs wild type (Fibro-WT) and NRVFs with LMNA D192G mutation (Fibro-MT) (nFibro-CT = 61, nFibro-WT = 63, nFibro-MT = 56 analyzed curves, respectively). B) Cell-Cell adhesion energy defined as the energy to separate two NRVFs. (nFibro-CT = 152, nFibro-WT = 221, nFibro-MT = 177 analyzed curves, respectively). C) Comparison of Maximum Force in cell-cell adhesion/de-adhesion expresses as the maximum force applied to separate two NRVFs, (nFibro-CT = 152, nFibro-WT = 221, nFibro-MT = 177 analyzed curves, respectively), (*** P < 0.0001). Numerical data is divided into quartiles, and a box is drawn between the first and third quartiles, with an additional line drawn along the second quartile to mark the median. The minimums and maximums outside the first and third quartiles are depicted with lines.
Round 3
Reviewer 2 Report
The manuscript can be accepted based on changes introduced by the authors